Evidence of phenotypic plasticity along an altitudinal gradient in the dung beetle Onthophagus proteus

Stanbrook Roisin A. roisin.stanbrook@ucf.edu 1
Harris W. Edwin 2
Wheater Charles P. 3
Jones Martin 3
1 Department of Biology, University of Central Florida , Orlando , FL , United States of America
2 Crop and Environment Sciences, Harper Adams University , Newport , United Kingdom
3 Department of Conservation and Ecology, The Manchester Metropolitan University , Manchester , Lancashire , United Kingdom
Andrew Nigel
Electronic publication date: 2021 Feb 24
Publication date: 2021
Volume: 9
Electronic Location ID: e10798
Received 2020 May 29; Accepted 2020 Dec 28
Copyright: ©2021 Stanbrook et al.
Copyright year: 2021
Copyright holder: Stanbrook et al.
License: This is an open access article distributed under the terms of the Creative Commons Attribution License, which permits unrestricted use, distribution, reproduction and adaptation in any medium and for any purpose provided that it is properly attributed. For attribution, the original author(s), title, publication source (PeerJ) and either DOI or URL of the article must be cited.
License URL: https://creativecommons.org/licenses/by/4.0/

Keywords: Polyphenism, Trait variation, Dung beetle, Altitude, Afromontane forest

Funding: The authors received no funding for this work.

==============================
Background

High altitude insects are an ecologically specialized group and possess a suite of adaptions which allow persistence in the inhospitable conditions often associated with mountain tops. Changes in body coloration and reductions or increases in body size are thought to be examples of such adaptions. Melanic individuals, or individuals containing high levels of eumelanin, possess several traits which increase resistance to ultraviolet radiation and desiccation, while aiding thermoregulation. Trait variation is often observed in dung beetles and is associated with dimorphism and sexual selection. In this study, we identified trait changes which occur across an altitudinal gradient by measuring morphological color and body size traits in a montane insect.

Methods

Using standard digital photography and Image J, we examined individuals of Afromontane dung beetle Onthophagus proteus. Individuals were classified according to sex and color morph to identify intrasexual variance. Nine morphometric traits were measured per beetle to identify patterns of morphology across discrete 500 m altitude segments.

Results

The results of this study provide one of the first descriptions of trait changes associated with elevation in an African dung beetle. We suggest that color polymorphism in Onthophagus proteus might be at least partly driven by environmental factors as there is significantly increased melanism with increasing elevation and significant differences in color hues between altitude bands. We also suggest changes in horn length are density dependent, as we observed an increase in cephalic horn length at high elevations where O. proteus is the most abundant species.

Introduction

The existence of discrete morphs associated with environmental factors is regularly a feature of intraspecific variation throughout the animal kingdom. Environmental factors commonly influence patterns of morphological variation within natural populations and the study of intraspecific variability along elevation gradients represents an appropriate natural experiment to understand the response of organisms to environmental changes which occur over short spatial distances (Körner, 2007). Studies investigating how phenotypic traits change along latitudinal or elevational gradients can contribute to the prediction of species responses to climate change by using a space-for-time substitution (Menéndez & Gutiérrez, 1996; Hodkinson, 2005; Birkett, Blackburn & Menendez, 2017). Organisms deal with prevailing environmental conditions by one of two main approaches, namely by evolving new genetic adaptations or through phenotypic plasticity. High altitude insects are an ecologically specialized group (Mani, 1968), possessing a suite of adaptions which allow persistence in inhospitable conditions associated with inhabiting mountaintops. One such adaptive trait is melanism, defined as ‘the occurrence of variant(s) that is/are mostly or completely dark in pigmentation as intraspecific polymorphisms’ (True, 2003). Melanistic individuals have frequently been observed in montane environments and there is evidence of a positive relationship between increasing altitude and increased melanism (ClusellaTrullas, Van Wyk & Spotila, 2007). Melanic individuals are characterized by possessing increased resistance to ultraviolet radiation, desiccation, and thermoregulation capacity. There are several hypotheses which associate darker body coloration with environmental change. The thermal melanism hypothesis (TMH), (e.g., Watt, 1968; Kingsolver, 1987) posits that body color is a significant factor affecting body temperature as darker colored individuals can attain higher body temperatures compared to lighter colored individuals when exposed to direct sunlight. Therefore, darker colored individuals may be better adapted to cooler regions as they can be active for longer periods while feeding, mating or during oviposition. This is especially pertinent for insects who live in mountainous tropical regions where the difference between atmospheric temperature and objects exposed to direct sunshine can vary greatly (Mani, 1968). The photo-protection hypothesis (PPH) (Law et al., 2020) predicts an increase in cuticle darkness in insects due to the protection increased melanization provides against UV-B radiation. Increased melanization correlates with greater UV-B radiation exposure in butterflies and Drosphilia (Bastide et al., 2014; Katoh, Tatsuta & Tsuji, 2018) and higher levels of melanin are able to protect against harmful ultraviolet radiation (Delmore, Brennan & Orr, 2018). A general relationship between increasing altitude and color variation at elevated sites at various locations has been noted in other insect orders (Guerrucci & Voisin, 1988) along with studies examining comparisons of morphology at both lowland versus highland sites (Rajpurohit, Parkash & Ramniwas, 2008; Karl, Geister & Fischer, 2009). However, the range of studies in which changes in morphology are described along a continuous altitudinal gradient are limited and we believe this is the first documented example in an African dung beetle.

Body size in insects may also vary as a response to altitudinal change (Hodkinson, 2005) with body size reduction influenced by a number of complex environmental factors. Insects should have reduced flight performance at high altitudes because of reduced air temperatures and changes in the physical properties of air and may compensate for reduced air density by altering wing or body morphology.  Reductions in mean body size, with the increase in altitude, has been observed in several Carabidae genera along with a progressive general flattening and increase in the width of the body, associated with increase in the convexity of the elytra (Mani, 1968). Another widely described phenotypic phenomenon in many arthropods is alternative male morphologies, for example, in dung beetles belonging to the family Scarabaeidae Moczek Emlen 1999, (Emlen et al., 2005; Kishi, Takakura & Nishida, 2015). Like most other Onthophagine dung beetles only males exhibit cephalic horns. Due to their variation in size and morphology these horns are ideal characteristics for studying the origin and diversification of novel traits (Moczek, 2006). Cephalic horns are used as weapons by male dung beetles to guard tunnel entrances where female mates are present (Knell, 2011). Horn size in dung beetles is considered to be sexually selected trait (Pomfret & Knell, 2006a; Pomfret & Knell, 2006b) and is thought to relate to reliable signal of male quality being indicative a male dung beetle’s nutritional history and physiological condition in comparison to other traits.

Onthophagus proteus is a medium-sized Afrotropical dung beetle with a distribution strictly limited range to high elevations in Uganda (Nyeko, 2009), Tanzania and Kenya (Davis & Dewhurst, 1993). In Kenya, the known species distribution of O. proteus is restricted to Mt. Kenya and Aberdare National Park (ANP). O. proteus is the most abundant Scarabaeine dung beetle between 2,500–4,000 m asl in ANP, with a vertical distribution spanning the lowland forest to the moorland in the uppermost region of the Park (Stanbrook, 2018). This dung beetle was first described in D’Orbigny’s (1913) ‘Synopsis des Onthophagides d’Afrique’. Its protean characteristics were evidently apparent as notes contained within the description define the elytral and pronotal color as being “extremely variable”. Three color variants were originally described by D’Orbigny: green/bronze (pronotum/elytra), or brown/bronze or black/black (i.e., melanistic).

The aim of this study is to describe phenotypic trait variation along an altitudinal gradient of a regionally endemic Afromontane dung beetle. We investigate two hypotheses. Firstly, we predicted the number of darker bodied individuals would be observed with greater frequency at higher elevations due to the potential conferment of thermoregulatory benefits associated with living at harsh high-altitude environments. Secondly, we hypothesized that trait size in O. proteus would be positively correlated with elevational increases. Specifically, we examine if traits linked to the overall body size but particularly the elytra of O. proteus will increase with elevation. This phenomenon has been observed in previous studies which have related the effect of temperature on wing morphology, resulting in larger wings with reduced wing loadings at lower temperatures/ higher altitudes (Gilchrist & Huey, 2004; Bai et al., 2016)

Materials & Methods

Sample collection

Dung beetles were collected using pitfall traps baited with 50 g of elephant dung during June–July 2015, and February–March 2016 in the Aberdare National Park, Kenya (0.4167°S, 36.9500°E). Data were collected with the authorization and help of the Kenya Wildlife Service under research permit number NACOSTI/P/15/0573/3206. The Aberdare mountain is an elongated massif, running approximately north south, parallel to the direction of the Rift Valley. The highest peaks are Oldonyo Lesatima (4,000 m) in the north and Il Kinangop (3,906 m) in the south. Shuttle Radar Topography Mission (SRTM) raster data at one arc second resolution (30 m along the equator) were used to create a three-dimensional model of the Aberdare mountain range in ArcScene 10.4. Data were categorized along arbitrary interval breaks of 500 m to delineate elevation bands. Base height was adjusted from 0 to 3.28 m, and the pixel value (z value) changed from 0 to 1 to provide a hillshade aspect. The data were then ‘floated’ (Bajjali, 2018) to create an elevation model. A polygon of the ANP boundary was overlaid on the model to enable the viewer to see how altitude bands are located within the ANP (Fig. 1). Trapping for dung beetles was conducted using eight baited pitfall traps placed on a linear transect which were placed as centrally within each elevation band as was logistically possible and were located in both open and closed canopy areas. Traps were emptied and re-baited every 24 h over a four-day period to provide a total sample of 64 samples per site. Once collected, dung beetles were transferred into a 70% ethanol solution for preservation and identification.

Dung beetles were identified as belonging to O. proteus by the dissection of the male genitalia. The aedeagus was removed and heated in a 5% potassium hydroxide solution until internal structures were soft. The internal sac was drawn out by gently pulling the outer portion of the sac from the inside of the sclerotized capsule of the aedeagus. The sac was then rinsed with 70% ethyl alcohol. The structures were prepared on microscope slides in liquid glycerin. Preparations on microscope slides were labelled with an individual number and the corresponding number of the preserved specimen. Individual O. proteus from each elevation band were sorted according to sex by examining the ventral surface of the penultimate segment of the abdomen, which is medially compressed in males, but equal in females and then further sorted into observed color morphs (Fig. 2). Only male dung beetles exhibit horns in O. proteus.

Figure 1 Lateral view (A) and perspective view (B) of a 3D elevation model of the Aberdare National Park, Kenya. Elevation bands run from 1,888 to 4,000 m above sea level.

Each elevation band represents a defined sample location.The Aberdare National Park boundary is indicated by the black boundary line. 3D model created using ArcScene 10.4 and STRM data at a 30 m resolution.

Figure 2 The observed morphospecies in Onthophagus proteus.

The intraspecific variation in colour found within O. proteus (A) Green Pronota, Brown Elytra (GrBr); (B) Brown Pronota, Brown Elytra (BrBr); (C) Black Pronota, Brown Elytron (BlkBr); (D) Green Pronota, Black Elytra (GrBlk); (E) Black Pronota, Black Elytra (BlkBlk).

Dung beetle traits

Ninety-nine individuals 47 males and 52 females (20 individuals per elevation band, except Band 3, Supplementary information) of dry, preserved O. proteus were photographed at 16x magnification with a Nikon D3100 (effective pixel count of 14.2 megapixels) that was attached to Leica M165C microscope. Each image was captured in uncompressed .nef format (Nikon Electronic Format). Both the microscope and camera were placed inside a lightbox to control for fluctuating light conditions. A ForensiGraph™ grey and color standard (http://www.forensigraph.co.uk) was included in each photograph to allow RGB calibrations for the amount of Red (R), Green (G) and Blue (B) to be derived for each image. Measurements of grey standards were taken by drawing a box over the area of interest on the grey color standard, and then using the histogram function in ImageJ (Schneider, Rasband & Eliceiri, 2012) to determine the mean grey scale value and standard deviation for each channel (sensu Stevens et al., 2007). To measure the reflectance, the R, G and B values of the box were averaged over a set of adjacent pixels. The R, G and B values of the grey standards were then plotted and fitted to a curve for each RGB channel. Six 2 mm2 regions of interest (ROI) were identified to obtain RGB values for each individual, consisting of two pronotal and four elytral measurements (Fig. S1). To test for linearity, we used linear regression as suggested by Stevens et al. (2007) to assess the relationship between the RGB values in each ROI and the reflectance values contained in the grey standard. ROI were analyzed using the ‘RGB Measure’ plugin in Image J (Schneider, Rasband & Eliceiri, 2012) with RGB pronotal and elytral values for each individual expressed as a proportion of 255 bytes within the RedGreenBlue triplet. Component values are stored as integer numbers in the range 0 to 255. If all the components are at zero, the result is black; if all are at maximum (255), the result is the most representable white. For example, if the following RGB color model (R = 0, G = 0, B = 255) was converted to a hexadecimal string the resulting color would be a vivid royal blue, but if the same model with a blue component equaling 20 was used, the resulting color would be much darker hue, appearing almost black to the human eye. Individuals were categorized by observed pronotal color then by elytron color, resulting in a pseudonym for each color morph. Individuals with black elytra and pronota were initially categorized as ‘BlkBlk’ but further analysis revealed the black hue was in fact a very dark ‘blue-black’ which was undistinguishable with the human eye.

We chose nine morphological traits based their documented sensitivity to elevational change (Mani, 1968; Brühl, 1997; Bui, Ziegler & Bonkowski, 2020) or relevance to sexual selection Cook 1987 (Emlen, 1996; Liang, Shieh & Huang, 2008; Köhler, Samietz & Schielzeth, 2017)). These traits are described and illustrated in (Table S1, Fig. 2). To assess phenotypic variation as a response to elevational change, we measured elytron length and width, and thorax depth. In dung beetles, flightless species exhibit rounder, truncated elytra in comparison with flying species. This modification is hypothesized to be an adaption to reduce desiccation in arid environments (Scholtz, 2000) and in montane environments is theorized to be necessary to maintain flight function in lower air pressure (Kohlmann, Solís & Alvarado, 2019). Wing area relative to thorax size (wing loading) is also known to increase in many montane insects resulting in larger wings which reduce the velocity needed to sustain flight.

Positive allometry in sexually selected traits and other characteristics occurs in many species of horned Onthophagine dung beetles (Emlen, 2001). In many species, male horn length covaries with other body measurements such as total body length, and pronotal and cephalic area resulting in larger males exhibiting larger cephalic or pronotal horns. We measured horn length, head length and width, pronotum head and width, and total body length to identify the relationship between these characteristics and male horn length, and to investigate if they correlate with elevational change. To ensure continuity between individuals when measuring body size traits dung beetles were orientated according to positions described in Hernández, Monteiro & Favila (2011) who outlined optimal Cartesian coordinates such as the points of convergence of structures, the apices of processes or their corresponding endpoints to record dung beetle body size measurements. Trait measurements were taken using the line tool bar found in the AxioVision software package and recorded in micrometers (µm).

Data analyses

A Kruskal Wallis test with Bonferroni corrections were used to compare all morphological traits between individuals from different elevation bands. Chi-squared goodness of fit was used to identify whether beetle color morphs varied in frequency across all elevation bands. We used a Mann–Whitney U-test with Wilcoxon paired post-hoc tests to assess for morphological trait differences among males and females between elevation bands. Linear regression was used to assess the relationship between horn size and the proportion of red, green, and blue found in male pronota and elytra. Data analyses were conducted in R v3.6.1 (R Core Team, 2019).

Results

Differences in morphological variables between elevation bands

The results of this study provide one of the first descriptions of trait changes associated with elevation in an African dung beetle. We found both horn length (HL) and pronotal width (PW) to be significantly different in males between altitude bands, (HL: w = 27.28 df = 4, p = 0.0013; PW: w = 9.07 df = 4, p = 0.05). In post hoc comparisons, horn length in elevation bands 1–3 were significantly different from those in elevation bands 4–5 (Fig. 3). The median horn length for beetles in elevation bands 4–5 was greater compared to those in lower bands. There was also a difference (p = 0.032) between pronotal width in elevation bands 1 and 2 compared to pronotal widths in bands 3,4 and 5. No significant differences in pronotal length (PL), elytron length (EL) and width (EW), abdomen depth (AD) and body length (BL) between sex or elevation band were detected (ST2).

Figure 3 Boxplots showing male horn length (A) and male pronotum width (B) in Onthophagus proteus.

Male horn length (A) and male pronotum width (B) of Onthophagus proteus. The median values are indicated by colored horizontal lines; 25th and 75th percentiles as the top and bottom of the boxes. The small colored circles indicate the distribution of values per altitude band (B1–B5).

Sexual dimorphism between color morphs

There were no significant differences in total body length or total elytral length between males and females of any color morph (Table S3). However, there was a significant difference in median pronotal length between males and females who had both brown pronota and elytra between elevation bands (Mann–Whitney U = 29, p = 0.002), with males having significantly longer pronotal discs. Horn length was a significant predictor of the average proportion of blue found in male dung beetle elytra (r2 = 0.55, df = 45, p = 0.0006), with males with bigger horns having a lower blue component which results in darker wings when compared with beetles with horns less than 3,532 mm in length (Fig. 4). The proportion of either red or green found in elytra was not a significant predictor of horn length in males (red, r2 = 0.64, df = 45, p = 0.91, green, r2 = 0.62, df = 45, p = 0.07).

Figure 4 Horn length versus proportion of blue in male Onthophagus proteus.

Darker elytron are found in males with longer cepahlic horns. The trendline (solid red line) is the non-linear trend obtained using loess.

Differences in color morph frequency between elevation bands

We found an association between beetle color and elevational band (χ2 = 131.42, df = 16, p = 0.018), with a greater frequency of darker beetles found at higher altitudes compared with lower altitudes than expected by chance (Fig. 5). We found a strong positive association between elevation and GrBr color morphs (11.89% contribution) and elevation band 1 (1,888–2,000 m) and between the BlkBr (18.94% contribution) and the BlkBlk (12.36% contribution) color morphs in elevation band five (3,501–4,000 m). These cells contribute 43.19% to the total Chi-square score and accounted for most of the difference between expected and observed values.

Figure 5 The frequency of occurrence of morphotypes in each altitude band.

GrBlk, Green Pronotum with Black Elytra; BrGr, Green Pronotum with Brown Elytra; BrBr, Brown Pronotum with Brown Elytra; BrBlk, Brown Pronotum with Black Elytra; BlkBlk, Black Pronotum with Black Elytra.

Discussion

Here we report altitudinal clines in morphological and color traits from the dung beetle O. proteus. Color morph variability was most pronounced in elevation band four which contains the largest spatial extent (295.6 km2) of all elevation bands as the majority of the park lies between 3,000–3,500 m asl (Fig. 1). This band also contains the greatest habitat heterogeneity (Hagenia dominated forest, Bamboo forest, and Ericaceous moorland) and therefore many more associated ecotones of any elevation band in the ANP. This diversity of habitat types and transitions may account for high variation in beetle color. The dispersal ability of O. proteus is unknown, however, a similar-sized dung beetle Canthon luctuosus which is also diurnal and lives in forest, was found have a maximal dispersal distance of 504.7 m (Da Silva & Hernández, 2015). This indicates that it would be possible for O. proteus to disperse between adjacent altitude bands in search of food resources and potential mates.

O. proteus does not undergo any clinal change in overall body size with altitude. Body length, pronotum length and elytron length did not change either in males or in females for each color morph, but horn length and pronotum width did vary across the altitudinal gradient with male individuals exhibiting longer horns and wider pronotal discs at higher elevations. We found that horn size was the best overall morphometric classifier of elevational placement and melanic (Blk/Blk) males had longer horns in comparison with other males. Furthermore, we observed that green and brown morphs, were widespread at lower altitudes, and progressively disappeared with increasing elevation. Simultaneously, brown and black morphs, rare at low elevations, increased in frequency with altitude, with a particularly steep increase between 2,500 m and 3,500 m asl becoming the most common morph at high altitudes (Fig. 5). This suggests that color polymorphism might be at least partly driven by environmental factors as a similar trend in color polymorphism has been found in montane Chrysomelidae beetles (Mikhailov, 2008), grasshoppers (Köhler, Samietz & Schielzeth, 2017) and paper wasps (de Souza, Mayorquin & Sarmiento, 2020).

O. proteus is the most abundant Scarabaeinae dung beetle and has the widest elevational range of any sampled Scarabaeinae dung beetle within the ANP (Stanbrook, 2018). Many species demonstrate a positive relationship between the altitudinal range over which species occur and elevation, and this relationship has been termed elevational Rapoport’s rule (Stevens, 1992). One of the conditions of Rapoport’s rule is based on the breadth of climatic conditions organisms experience along gradients (Stevens, 1992). The evolution of intraspecific polymorphic traits along elevational gradients may allow species to adapt under variable and often harsh climatic conditions which can be indicative of montane environments. These adaptions may occur as directional changes in morphological traits, such as wingspan reductions or enlargement (Hodkinson, 2005; Eweleit & Reinhold, 2014; McCulloch & Waters, 2018) or color polymorphism which is considered an adaptive trait and is beneficial in aiding thermoregulation and decreasing ultraviolet penetration (Schweiger & Beierkuhnlein, 2016).

The color variation O. proteus might also be at least partly driven by the heightened levels of exposure to ultraviolet radiation. The ANP has a peak elevation of 4,000 m asl and is situated almost directly on the equator. This equatorial location means that the ultraviolet index for that particular area of Kenya can reach up to 14 at midday which is high when compared with upland areas at higher latitudes or the ultraviolet intensity at sea level. Traits which confer protection against increased UV penetration may be selected for in equatorial montane populations while beetles in lower altitudes tend to display an array of bright colors that is thought to play a vital role as an anti-predatory strategy (Tan et al., 2017). For example, green and brown coloration provides excellent camouflage against the densely vegetated backgrounds that tend to be more prevalent in the lower altitudes of montane environments, and vegetation in general.

Variation in trait size in Onthophagine dung beetles may be influenced by other factors, including maternal effects. Buzatto, Tomkins & Simmons (2012) explored the mechanism by which female dung beetles adaptively respond to perceived increased population density by preparing their male offspring for the level of sexual competition they will face as adults with larger horns. They discovered major male offspring of the same body size could have significantly larger horn if they were produced by females that experienced high population density than if they were produced by females that experienced low population density. We posit that the increased prevalence of male dung beetles with longer horns at higher altitudes may be an example of density dependent polyphenism. Density dependent polyphenism in males as a result of mean crowding has been described before in African Onthophagine dung beetles (Pomfret & Knell, 2008). O. proteus is abundant throughout all elevation clines in the ANP but is the most abundant dung beetle at the highest elevations in the Aberdare National Park (Stanbrook et al., in press). This abundance may potentially explain the large variation in horn length at upper elevations in the ANP as female perception of increased population density may result in major male offspring being produced with larger horns.

Conclusions

We describe altitudinal trends in morphology and body coloration of the dung beetle O. proteus. Our results are consistent with those of other studies which have found intraspecific color variation in insects along an elevation gradient. Our prediction that the number of darker bodied individuals would be observed with greater frequency at higher elevations due to the potential conferment of thermoregulatory benefits associated with living at harsh high-altitude environments held true as we identified an association between beetle color and elevational band, confirming a greater frequency of darker beetles are found at higher altitudes in the ANP.

We also hypothesized that trait size in O. proteus would be positively correlated with elevational increases. While high-altitude populations of O. proteus do not show an overall reduction in body size, male horn length and pronotal width were found to be significantly different between altitudinal bands. Both color and trait size adaptions may be explained by an increased need for thermoregulatory capacity under restrictive conditions at high altitudes which may help to maintain a longer activity window to find mates or limited food resources and thereby permit survival at high altitudes.

Supplemental Information

Supplemental Information 1 Approximate sample location of each RGB measurement

Click here for additional data file.

Supplemental Information 2 Approximate sample location of each morphometric trait within Onthophagus proteus

Click here for additional data file.

Supplemental Information 3 Description of each morphological trait used in the analysis

Click here for additional data file.

Supplemental Information 4 Statistics and p-values for non-significant results

The results of Kruskal Wallis test with Bonferroni corrections (H) used to compare morphological traits between individuals from different elevation bands and the results of Mann Whitney U-tests (w) with paired post-hoc tests to assess for morphological trait differences between sex. P values <0.05 are statistically significant.

Click here for additional data file.

Supplemental Information 5 Differences in total Elytron and Body length between male and female dung beetle color morphs

Click here for additional data file.

Supplemental Information 6 Raw data

Click here for additional data file.

Data were collected with the authorization and help of the Kenya Wildlife Service. We also thank Dr. Dmitri Logunov at the Manchester Museum for the use of their photographic facilities.

Additional Information and Declarations

Competing Interests

Author Contributions

Field Study Permissions

Data Availability

The authors declare there are no competing interests.

Roisin A. Stanbrook conceived and designed the experiments, performed the experiments, analyzed the data, prepared figures and/or tables, authored or reviewed drafts of the paper, and approved the final draft.

W. Edwin Harris and Martin Jones conceived and designed the experiments, authored or reviewed drafts of the paper, and approved the final draft.

Charles P. Wheater conceived and designed the experiments, analyzed the data, authored or reviewed drafts of the paper, and approved the final draft.

The following information was supplied relating to field study approvals (i.e., approving body and any reference numbers):

Data were collected with the authorization and help of the Kenya Wildlife Service under research permit number NACOSTI/P/15/0573/3206.

The following information was supplied regarding data availability:

The raw color and body size measurements for all dung beetles used in statistical analyses are available in the Supplemental Files.

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
