# Peer review of "Evidence of phenotypic plasticity along an altitudinal gradient in the dung beetle Onthophagus proteus"

_PeerJ, doi:10.7717/peerj.10798_

## Round 0.1 · original submission · Major Revisions

This is a nice study, and both reviewers see merit in the research. Both also require you to address some issues, particularly Reviewer 2.

In addition, I also have some queries:

Line 65 – provide a textbook example here
Line 143 – what bait (and how much) did you use as an attractant? More details on trap type and time spent trapping required.
Line 188 – for the traits that you studied it would also be good for you to identify why you assessed these particular traits – what ecological reason is it important to use these. I would like to see an overview provided here, rather than just referral to another paper: as is it is quite central to your argument.
I really think it would be beneficial for you to also provide your r-code here, as well as the raw data, so readers can run the analysis. Not only to be fully open – especially when using a new technique – I think it will also make your paper more cited, as people can see how you went about your analysis, and can replicate it.
Line 381 – I would think that these species are regional endotherms, rather than ectotherms. Dung beetles can regulate their temperature internally, see: Verdú J. R., Arellano L. & Numa C. (2006) Thermoregulation in endothermic dung beetles (Coleoptera: Scarabaeidae): Effect of body size and ecophysiological constraints in flight. J Ins Phys 52, 854-60.
Heinrich B. & Bartholomew G. A. (1979) Roles of endothermy and size in inter- and intraspecific competition for elephant dung in an african dung beetle, Scarabaeus laevistriatus. Physiol. Zool. 52, 484-96.
Also, clean up your reference list – italicise species names; consistent capitalisation; correct author determination (e.g. line 618); remove publishers of journals (e.g. line 629); not ALL CAPS (line 639). It provides a good overview of how well you have read and reviewed your full manuscript.
Table 1. I have no idea what this table is showing, or how to interpret it. I think you need some explanatory text to tell the reader what is being presented here.

·

Basic reporting

The manuscript reports changes in morphological and color traits in an insect along an elevational cline. By investigating a conspicuous polymorphic species whose coloration was not previously studied in detail, plus the combination of a modern statistical approach, I believe that this manuscript has merit to be published on Peer J. Overall, it is well written, well structured, figures are relevant, high quality and well-described. I enjoyed reading it!

Experimental design

The study is well-designed, well-detailed and it nearly contains all the necessary information to allow its replication.

Validity of the findings

Conclusions are well-supported.

Additional comments

I just have very few minor suggestions, bellow:

• There are some very recent and exciting literature on changes in insect body color according to elevation. For example, see De Souza et al. (2020) and references therein. This (and other) studies pretty match the findings reported to O. proteus, and I believe it is important to mention that the pattern reported to this beetle is quite common. This can allow authors to better connect their findings to the hole body of evidence on this topic. I suggest authors to mention at least some of this literature. (de Souza, A. R., Mayorquin, A. Z., & Sarmiento, C. E. (2020). Paper wasps are darker at high elevation. Journal of Thermal Biology, 89, 102535.)

• The methods are overall well-described, but I suggest authors to include a bit more information about samplings, to allow replication. For example, how many traps were used? How long the traps stayed in the field? at the time of taking pictures, were the insects alive or died? How did you keep them between collection and taking the pictures (ethanol, frozen, dried)?

• Line 365, change “Onthophagus proteus” to “O. proteus”

• Line 331, I guess it is “pronota” instead of “pronata”

Reviewer 2 ·

Basic reporting

At it's core, this is a nice, simple study testing for associations between color, morphology, and elevation in an African dung beetle based on specimens collected in the field. Unfortunatley, this releatively straightforward premise is obscured by extremely wordy presentation in most sections of manuscript, with paragraphs dedicated to topics with only tangential relevance to the study at hand. For example, an entire paragraph (lines 84-91) is dedicated to plasticity without connecting it to any of the study questions, and this is not a study of plasticity regardless. At best plasticity would be brought up in the discussion while addressing the potential mechanisms underlying the observed patterns, but this topic is not touched on at all there. Another example is lines 403-424 in the discussion, a long paragraph dedicated to introducing horn dimorphism and sexual selection in dung beetles without direct connection to this study. Both of these paragraphs could be entirely dropped with little loss to the manuscript. These are just a few of many examples, and more are pointed out in the annotated manuscript attached.

Experimental design

The core experimental question and objective is nicely stated at the end of the introduction (lines 115-126). The methods themselves are nicely explained and illustrated using supplemental figures. I am overall very impressed by the quality of the photographic methods and their reporting, a subject I often find under-reported elsewhere. Only a few details are missing to ensure this manuscript would be entirely replicable. I would like to know 1) what kind of bait was used in the traps, 2) how the gray values were used for calibration, and 3) whether the color values were appropriately linearized (see Stevens et al. 2007, cited by the authors, for more details on the photographic aspects).

Validity of the findings

A large portion of the analysis comes from a Conditional inference tree (CIT) approach. I will admit, I am not familiar with this method having not encountered it before, and thus am not in position to evaluate whether it was conducted correctly. That said, I do question whether it is necessary for this study. The core results are entirely provided by the more traditional statistical tests already conducted in the manuscript. The CIT mostly confirms these results (horn morphology and color are associated with elevation). I do not see what is the added value of being able to predict altitudinal band of a particular specimen, particularly because these bands are an artificial grouping of a continuious variable (altitude). Prediction of phenotype based on altitude seems like a more useful goal because it is more in line with the primary hypothesis of the study (that morphology and color are associated with elevation), and these predictions can already be made by using the traditional statistical results.

The early discussion (327-362) focuses on trying to explain the CIT results which creates a much more confusing and complicated presentation of overall straightforward results. I acknowledge that substantial text is dedicated to explaining the broad utility and potential of CIT methods (208-244, 474-479), but how they fit into and contribute to this study is not at all clear. As such, I recommend dropping the CIT analysis unless the authors can clearly justify the added value of the CIT approach for addressing their hypotheses and ensure that the discussion remains clear and concise despite this. As is, the use of the CIT approach feels like novel statistics just for the sake of using them, without adding meaningfully to the results of this study, and indeed it makes the results harder to understand.

Additionally, non-significant statistical tests were conducted, but were left out of the manuscript (for example, the relationship between morphological traits other than horn-length and elevation). These need to be included at least in a supplemental table, but ideally in the main paper. Non-significant does not mean nothing is happening, just that you don't have support for something happening. The meaning of a non-significant p value is very different between p=0.9 and p=0.09, so these need to be provided.

Raw data is provided, and seems mostly sufficient, except that RGB_Ely_G and RGB_Ely_R values are missing for some specimens, without any clear explanation of why they are missing.

Additional comments

Overall, I think there is a nice, simple study at the core of this manuscript. Unfortuantely, that straightforward core (dark morphs are more common at higher elevations and they also have longer horns at longer elevations) is lost in an excess of extraneous information and needlessly complex statistical methods. I have left specific comments in the annotated manuscript which will hopefully help in focusing the manuscript and also help make other small improvements.

Annotated reviews are not available for download in order to protect the identity of reviewers who chose to remain anonymous.

---

## Round 0.2 · Minor Revisions

The revised manuscript is much improved. The reviewer that it was returned to has some issues that need to be resolved/acknowledged before acceptance. Please revise/rebut accordingly.

Reviewer 2 ·

Basic reporting

I got to review the previous version of this manuscript, and I appreciate how, as requested, substantial extraneous information has been removed greatly improving the overall quality and readability of the manuscript. That said, some information remains in the discussion that is unconnected to the study itself. Specifically, the paragraphs at lines 408-417 and 431-440 should be either dropped or their connection to the study's results should be made clear (either would be fine).

The raw data is now good, and the R script is well annotated.

Experimental design

The reporting of the experimental design and methods was solid in the original manuscript with only a few missing details. Those holes have now been filled in and the level of detail on the methods is now excellent.

Validity of the findings

My primary complaint regarding the previous version of this manuscript was the use of conditional inference trees (CIT) for much of the analysis and the heavy emphasis placed on this approach, which I believed introduced a great deal of complexity to the analysis while adding very little to the results.

The authors have written an extensive reply to my critique, and while I now better understand the potential merits of CITs (such as avoiding many common statistical assumptions), I find my skepticism of their use in this specific context remains (little of the response to reviewers addresses why to use CIT in this specific study). I believe I can more precisely articulate my issues now.

My main issue with the CIT analysis here is that the study's hypotheses are that morphology and color will change with elevation. The CIT, on the other hand examines how elevation changes with morphology and color, inverting the response and predictor variables. (I understand this data is correlative, but there is still directionality implied by the hypotheses that needs to be kept in mind.) CIT also seeks to make predictions rather than test a hypothesis (yes these are different, if related, goals). This obscures how the results connect to the hypothesis and greatly increases the complexity of an otherwise straightforward study, but it also raises greater problems.

The main problem arising from this backwards approach to these hypotheses is that by trying to predict elevation from traits, this approach only highlights the strongest elevation-trait relationships. Your hypotheses are not about which traits have the strongest relationship to elevation. They are about whether different traits are related to elevation, independently of each other. Just because one or a few traits are sufficient to predict altitude, it doesn't mean others don't also change. Multiple traits may be strongly correlated and thus add little predictive power for elevation, but if we care whether each trait changes with elevation (and based on this study's hypotheses we do care) then we won't learn about these other relationships from the CIT.

You argue that CITs avoid many of the assumptions of parametric methods a(although note that your other analyses are mostly non-parametric) and can incorporate complex interactions allowing them to make better predictions than other methods. None of this matters if it is testing the wrong hypothesis. Multiple regression to predict elevation from all the different traits would also have little value in this context. If you really wanted to incorporate all the different traits into a single analysis, you could use a multivariate approach to the response variables (such as PCA), but I think this study is fine as is testing the different traits independently and interpreting the set of those that change.

The CIT analysis could be dropped from this manuscript altogether and little of meaning would be lost (except, perhaps, evidence that some of the conclusions are supported by multiple statistical approaches in a convoluted to interpret way). At minimum, its role and priority in the discussion should be reduced and the conclusions should not focus on it (as is the conclusions are more than half about CITs, rather than the results of the study itself). You also state at line 360 that the full model (model C) was the best one, but so far as I can tell from the results, model B (just morphology) was actually better.

Alternatively you could rephrase your hypotheses in the introduction to be about which traits will be most useful for predicting what elevation beetles will be found at. I don't think this is a good idea.

As a final note, I think there remains some non-significant results that don't seem to be reported in the manuscript or supplemental materials (specifically those from the paragraph at line 290) and you should include the actual magnitude of difference through summary statistics or plots for some of your other significant results (lines 281-286).

Additional comments

I appreciate the effort the authors took to address my comments on the previous version of this manuscript. I find it generally improved (although see the annotated manuscript for more changes. Most are grammatical but a few additional details or connections to the hypotheses are requested). That said, I am still not convinced regarding the use of CITs in this specific study and the heavy emphasis placed on those results in the discussion (as opposed to the more traditional non-parametric tests already used). At minimum, their role in the conclusion and primacy in the discussion need to be reduced.

Annotated reviews are not available for download in order to protect the identity of reviewers who chose to remain anonymous.

---

## Round 0.3 · accepted · Accept

Dear Roisin and co-authors, thank you for the revisions made to your manuscript after the second round of peer review. In my opinion, the paper is now acceptable, for publication. I note that you removed the conditional inference trees (CIT) analysis and relevant statistical code - I think this was wise as it really does make the work much clearer and easier to follow, Although I suspect many hours went into the analysis - and it would have been a little painful to remove the stats and associated code - it may be useful to develop a separate paper using CIT to explain its use and benefits in more detail - e.g. as a specific methods paper!

Looking forward to seeing the work published. Merry Christmas!
Nige